# Research on a U-Net Bridge Crack Identification and Feature-Calculation Methods Based on a CBAM Attention Mechanism

**Huifeng Su [1,\*], Xiang Wang [1], Tao Han [2], Ziyi Wang [1], Zhongxiao Zhao [1] and Pengfei Zhang [1]**

1   College of Transportation, Shandong University of Science and Technology, Qingdao 266590, China
2   Shandong Expressway Qingdao Development Co., Ltd., Qingdao 266000, China
*   Correspondence: skd991970@sdust.edu.cn

**Abstract:** Crack detection on bridges is an important part of assessing whether a bridge is safe for service. The methods using manual inspection and bridge-inspection vehicles have disadvantages, such as low efficiency and affecting road traffic. We have conducted an in-depth study of bridge-crack detection methods and have proposed a bridge crack identification algorithm for Unet, called the CBAM-Unet algorithm. CBAM (Convolutional Block Attention Module) is a lightweight convolutional attention module that combines a channel attention module (CAM) and a spatial attention module (SAM), which use an attention mechanism on a channel and spatially, respectively. CBAM takes into account the characteristics of bridge cracks. When the attention mechanism is used, the ability to express shallow feature information is enhanced, making the identified cracks more complete and accurate. Experimental results show that the algorithm can achieve an accuracy of 92.66% for crack identification. We used Gaussian fuzzy, Otsu and medial skeletonization algorithms to realise the post-processing of an image and obtain a medial skeleton map. A crack feature measurement algorithm based on the skeletonised image is proposed, which completes the measurement of the maximum width and length of the crack with errors of 1–6% and 1–8%, respectively, meeting the detection standard. The bridge crack feature extraction algorithm we present, CBAM-Unet, can effectively complete the crack-identification task, and the obtained image segmentation accuracy and parameter calculation meet the standards and requirements. This method greatly improves detection efficiency and accuracy, reduces detection costs and improves detection efficiency.

**Keywords:** U-net; attention mechanism; bridge crack; crack feature measurement

## 1. Introduction

Crack detection and prevention are critical for bridge maintenance, and the demand for crack detection increases as the service life of a bridge increases [1]. If cracks can be detected at an early stage and their development can be tracked in real-time, the maintenance costs of the bridges will be greatly reduced, and the safety of traffic will also be ensured. Concrete structures are commonly used in bridge construction, as they are made from a wide range of materials, are relatively inexpensive, and have high strength, durability and plasticity to meet the needs of construction projects. However, due to environmental factors, their shrinkage, and other factors, the bridge itself is subject to varying degrees of disease. Cracking is one of the most common forms of concrete bridge disease and is also the most harmful. Reinforced concrete bridge defects can take many forms and occur in a variety of locations. Any damage to the concrete structure generally begins with cracks in the concrete, and adverse weather conditions can cause rainwater to penetrate the cracks and come into contact with the reinforcements, resulting in varying degrees of corrosion of the reinforcements, leading to a shortened service life and increased risk. It is therefore of great importance to carry out regular inspection and health assessments of bridges and to maintain and strengthen them following the assessment results to ensure that cracks are

detected and repaired at the earliest stage, to reduce the risk factor significantly, reduce the capital investment and prevent the problems before they occur.

Traditional bridge crack detection is mainly based on manual measurements, which are inefficient, have a high miss rate and are time-consuming and costly. Furthermore, the calculation and processing of parameters such as crack width and length are relatively slow. Therefore, automatic and efficient crack detection is essential for bridge structural health assessments [2–9].

In recent years, image-based crack detection has received increasing attention in the field of non-destructive testing. The main advantage of image-based crack detection is the use of image processing methods [10], which can provide accurate results compared to traditional manual detection methods. Deep convolutional neural networks (DCNN) have been shown to be competitive with and sometimes superior to humans in solving many computer vision problems, such as image recognition [11], target detection [12] and semantic image segmentation [13]. For different cases of cracks, scholars have proposed a series of neural network-based crack detection and recognition algorithms. Eisenbach et al. [14] proposed a road disease dataset for training deep learning networks and provided the first evaluation of the latest techniques for road disease detection. Xu et al. [15] proposed a Faster R-CNN and Mask R-CNN joint training strategy that can achieve better results than YOLOv3 with few training images. Fan et al. [16] proposed a deep-learning-based supervised method. The method can effectively detect pavement cracks in different environments by varying the positive and negative ratios of the samples. Long J. et al. [17] were the first to propose a fully convolutional network (FCN) for semantic image segmentation, and various semantic segmentation networks have subsequently emerged. Among them, Ronneberger et al. [18] proposed a U-net semantic segmentation network for biomedical image segmentation, which has a multidimensional feature in the upper sampling layer. The shallow layer is used to solve the pixel localization problem, while the deep layer is used to solve the pixel classification problem. It requires only one training session to complete an image segmentation task. Unlike CNNs, the biomedical image segmentation results are more accurate when U-net contains fewer training images. Conventional U-net neural network structures generally contain both encoder and decoder components, and both are built from similar modules [19]. Liu [20] used a U-net network to detect concrete cracks and found that the trained U-net could identify the crack location from the input original image under various conditions, with high effectiveness and robustness. Shankaranarayana et al. [21] combined the residual module and U-net to propose Res-Unet, which deepened the network structure, enhanced the fusion of shallow and deep features and accelerated the convergence of the network while avoiding the gradient disappearance and explosion problems. Oktay et al. [22] applied the attention mechanism to the U-net segmentation network and proposed a U-net based on attention, which can better capture prominent features and suppress irrelevant background areas. Fan et al. [23] proposed an improved U-net remote sensing classification algorithm that fuses attention and multiscale features. It connects spatial pyramidal pooling with the convolutional units of the original U-net in the form of residuals, enhances the expression of shallow features and uses a spatial attention mechanism to combine spatial information with semantic information, enabling the decoder to recover more spatial information. Ma [24] proposed joint-attention feature fusion to enhance object detection performance. To exploit dependencies, channel attention and position attention modules are used for different scales, which are implanted in order.

The above work provides implications for the research and development of crack image segmentation. In order to enhance the segmentation capability of the network and improve the segmentation effect of the detailed regions of an image, we applied the semantic segmentation idea to multiscale concrete crack detection and propose a concrete crack detection method based on the U-net model (CBAM-Unet) with a two-channel attention mechanism with channel attention and spatial attention. Figure 1 shows the general flowchart for extracting crack features based on CBAM-Unet. The addition of

channel attention and spatial attention to each max-pooling process increases the integrity of feature extraction during the max-pooling process, thereby making the training results more accurate and improving the accuracy of multi-scale crack identification. Using on the CBAM-Unet algorithm to complete the crack image segmentation task, the crack image is post-processed with digital image processing methods to extract the crack morphology, length and width information with specific steps, including binarization, Gaussian blurring, Otsu threshold segmentation, crack skeletonization, crack length, width measurement, etc. The experimental results demonstrate that the CBAM-Unet model further improves the segmentation capability of complex bridge cracks compared to the original U-net algorithm, and the method used in this paper can accurately extract information such as crack length and maximum width.

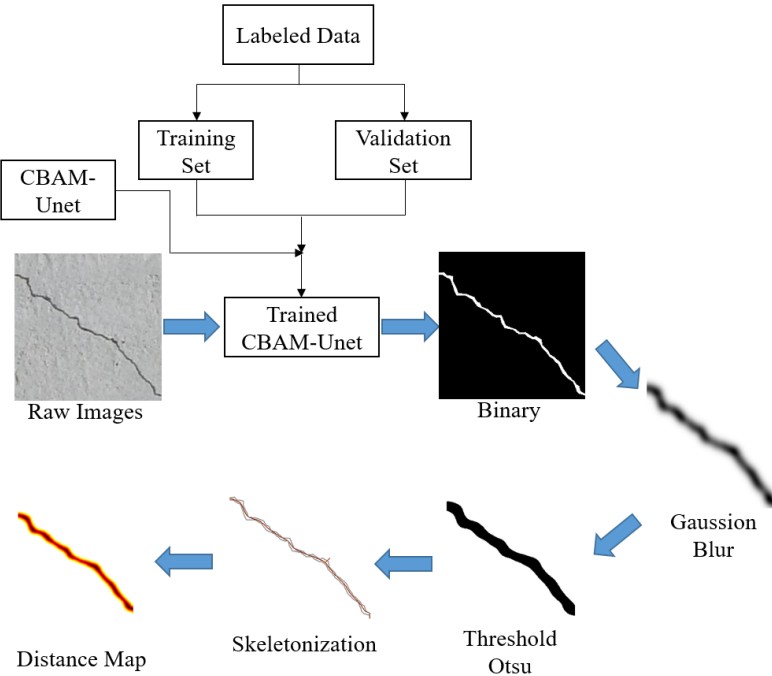

**Figure 1.** Extraction of fracture features based on CBAM-Unet.

## 2. U-Net Methods and Channels, Spatial Attention Mechanisms

### 2.1. U-Net

U-net is a U-shaped symmetrical structured network. The network consists of two parts, a compressed path and an extended path. The compressed path consists of four coding blocks, each of which contains two convolutional layers and a maximum pooling layer, and the convolutional layer extends the channel. The maximum pooling layer compresses the size of the feature map, reducing the size of the feature map to half of the original size and doubling the number of feature maps after each coding block. Each decoding block contains two convolutional layers and one deconvolutional layer. The convolutional layer compresses the channel, and the deconvolutional layer restores the feature map size, expanding the feature map to twice the original size and reducing the number of feature maps by half for each encoding block. In the jump–join stage, the output of each encoding block is spliced with the input feature map of the decoding block of the same level to recover some of the semantic information lost during the encoding process, thereby ensuring the accuracy of the segmentation. Finally, the number of channels is compressed to the number of classifications performed. The U-net model structure is shown in Figure 2.

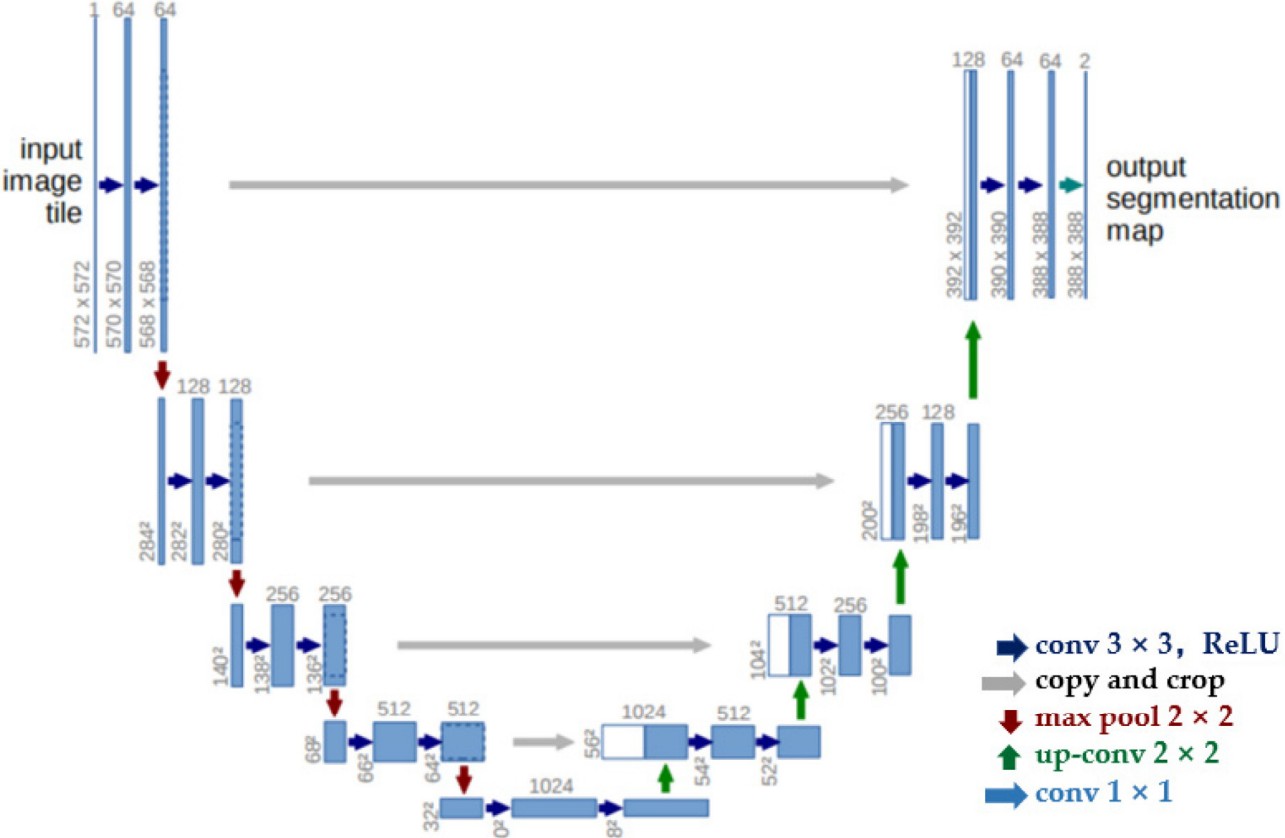

**Figure 2.** U-net structure.

### 2.2. Design of CBAM-Unet Based on an Attention Mechanism

In computer vision, the ability to focus attention on important areas of an image and discard irrelevant ones is known as an attention mechanism. In the human visual cortex, attention mechanisms are used to analyse complex scene information quickly and efficiently. This mechanism was later introduced into computer vision to improve performance. Attention mechanisms can be thought of as dynamic selection processes for important information from an image, which is achieved by the adaptive weighting of features. Attention mechanisms have greatly improved the performance of computer vision tasks. An attention mechanism in the field of deep learning filters irrelevant information from a large number of deep learning samples and selects information that is more critical to the current task, and is widely used in various types of tasks, such as natural language processing, image recognition and speech recognition.

The structure of the proposed algorithm is shown in Figure 3. The improved model still uses the encoder–decoder architecture and adds a CBAM module to the encoder network for adaptive feature refinement of the input feature map at each max-pooling layer.

The convolutional block attention module (CBAM) used in this paper is a lightweight attention module. Given a feature map, the CBAM module can serially generate attention feature map information in both channel and spatial dimensions, and then multiply the two sets of feature map information with the original input feature map for adaptive feature correction to produce the final feature map [25–27].

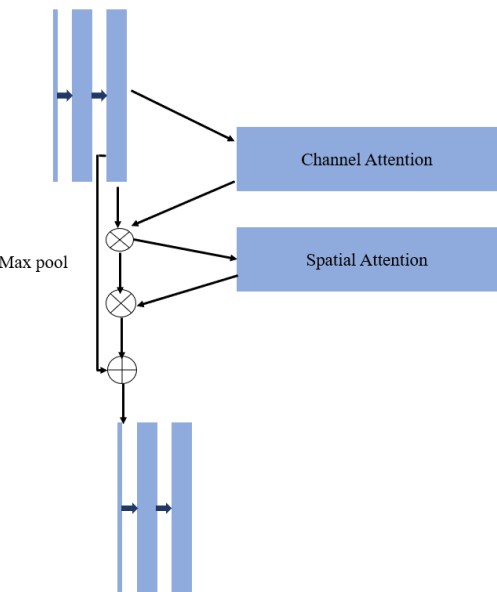

**Figure 3.** Improved partial algorithm model structure.

The overall process of CBAM is as follows:

Given an intermediate feature map ($F \in R^{C \times H \times W}$) as input, the overall CBAM operation process is divided into two parts—firstly, global maximum pooling and average pooling of the input by channel, and the two pooled one-dimensional vectors are fed into the fully connected layer operation and then summed to generate one-dimensional channel attention ($M_C \in R^{C \times 1 \times 1}$). Then, it multiplies the channel attention with the input elements to obtain the channel attention adjusted feature map $F'$. Global maximum pooling and average pooling of $F'$ by space are performed. The two pooled two-dimensional vectors are stitched together and then convolved to finally generate a two-dimensional spatial attention ($M_S \in R^{1 \times H \times W}$). Then, it multiplies the spatial attention with $F'$ by elements. The exact process is shown in the diagram above. The CBAM process of generating attention can be described as the following equation:

$$F' = M_C(F) \otimes F, \tag{1}$$

$$F'' = M_S(F') \otimes F', \tag{2}$$

where $\otimes$ represents element-level multiplication with a broadcast mechanism for dimensional transformation and matching in between.

The max-pooling process of the encoding part is first performed in the channel attention module [28]; the input feature map size is $H \times W \times C$, and two $1 \times 1 \times C$ feature maps are first obtained by MaxPool and AvgPool, respectively. Then, these two feature maps are fed into the two fully connected layers. Then, the two feature maps are added together, and then the weight coefficients between 0 and 1 are obtained by the sigmoid function. Then, the weight coefficients are multiplied with the input feature map to obtain the final output feature map.

The formula for the channel attention module is as follows.

$$M_C(F) = \sigma(MLP(AvgPool(F)) + MLP(MaxPool(F))) = \sigma(W_1(W_0(F_{avg}^C)) + W_1(W_0(F_{max}^C))) \tag{3}$$

In the above equation, MLP (Multilayer Perceptron) represents the shared MLP module in the channel attention module. In this module, the number of channels is first compressed and then extended to the original number of channels. The result of the two activations is obtained by the ReLU activation function.

After the output of the channel attention module, the spatial attention module is then introduced to focus on which part of the space has meaningful features [29]; the input is

H × W × C. The maximum pooling and average pooling of one channel dimension are performed to obtain two feature maps of H × W × 1, and then the two feature maps are stitched together in the channel. Next, the two feature maps are stitched together in the channel dimension, and now the feature map is H × W × 2. Then, after a convolution layer, it is reduced to 1 channel, and the convolution kernel is 7 × 7. While keeping HW unchanged, the output feature map is H × W × 1. The final feature map is obtained by multiplying the sigmoid function with the input feature map.

The formula for the spatial attention module is as follows.

$$M_S(F) = \sigma(f^{7\times7}([\text{AvgPool}(F); \text{MaxPool}(F)])) = \sigma(f^{7\times7}(F_{avg}^S; F_{max}^S) \qquad (4)$$

The structure of the CBAM module is shown in Figure 4a–c.

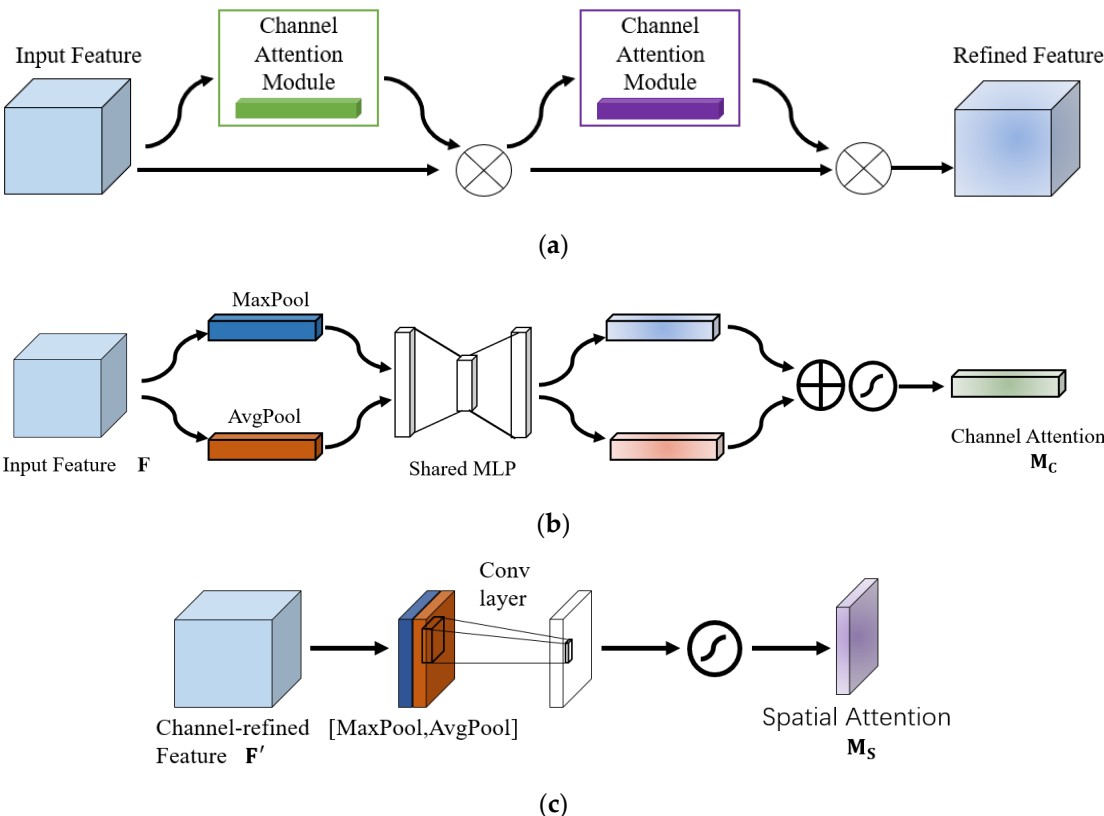

**Figure 4.** CBAM and internal structure: (**a**) CBAM module structure; (**b**) channel attention structure; (**c**) spatial attention structure.

## 3. Crack Geometry Measurement Algorithm

In the actual bridge-crack-detection process, the information on cracks includes information such as the maximum width and length of the crack, among which the measurement of the maximum crack width value is the most important point in the actual inspection. For regular cracks, the crack length needs to be calculated as a secondary reference in order to better describe the extent of the crack damage.

After CBAM-Unet image crack identification and localization are completed, the crack image is post-processed using digital image processing methods to extract crack morphology, length and width information, with specific steps such as binarization, Gaussian blurring, Otsu threshold segmentation, crack skeletonization, crack length and width measurement, as shown in Figure 5.

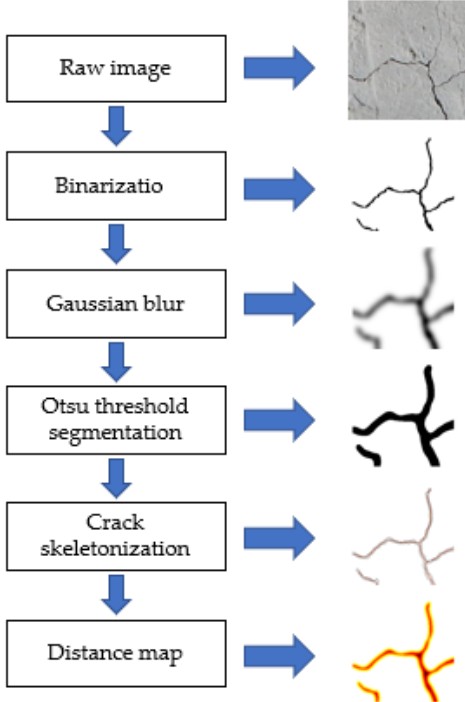

**Figure 5.** The process of digital image processing methods.

(1) Image binarization. The maximum inter-class variance method is used to convert each pixel of the grey-scale image to 0 or 255, reducing the number of image data and highlighting the target contours.

(2) Gaussian blur (GB) [30]. Blur in an image means that the pixel value of the central pixel is the average of the sum of the pixel values of the surrounding pixels, as shown in Figure 6.

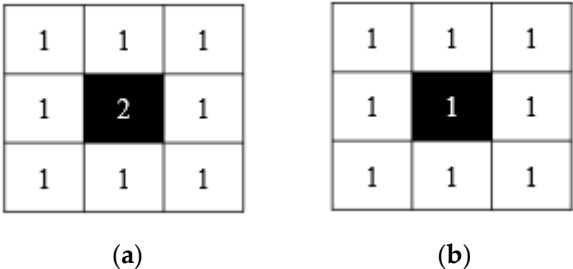

| (a) | (b) |

**Figure 6.** Gaussian Blur. (**a**) Raw image (**b**) after blurring.

The first image is the raw image with a pixel value of 2 for the central pixel, and the second image is the image after blurring the central pixel, with the pixel value being the average of the sum of the surrounding pixel values. The Gaussian blurring process uses a two-dimensional normal distribution, with the central pixel as the origin and the other points assigned weights according to their positions on the normal curve, to obtain a weighted average, and each point is multiplied by its own weight to obtain a weighted average of the Gaussian blurred value of the central point.

(3) Otsu threshold segmentation

Otsu is a method for automatically determining thresholds using the maximum inter-class variance [31–33]. It is a global binarization-based algorithm that divides an image into two parts, foreground and background, based on the grey-scale characteristics of the image. When the optimal threshold is taken, the difference between the two parts

should be the maximum, and the measure of difference used in the Otsu algorithm is the more common maximum interclass variance. The larger the interclass variance between foreground and background, the greater the difference between the two parts of the image. When part of the target is misclassified into the background or part of the background is misclassified into the target, the difference between the two parts becomes smaller, and when the segmentation of the chosen threshold maximises the interclass variance, it means that the probability of misclassification is minimal.

Set T to be the segmentation threshold of foreground and background. The proportion of foreground points in the image is $\omega_0$, and the average grey level is $\mu_0$; the proportion of background points in the image is $\omega_1$, and the average grey level is $\mu_1$; and the total average grey level of the image is $\mu$. For the variance g of the foreground and background images, the formulae are as follows.

$$\mu = \omega_0 \times \mu_0 + \omega_1 \times \mu_1 \tag{5}$$

$$g = \omega_0 \times (\mu_0 - \mu)^2 + \omega_1 \times (\mu_1 - \mu)^2 \tag{6}$$

The above two equations are combined:

$$g = \omega_0 \times \omega_1 \times (\mu_0 - \mu_1)^2 \tag{7}$$

When the variance g is maximum, it can be assumed that the difference between foreground and background is greatest at this point, and the grey level T at this point is the optimal threshold. The interclass variance method is very sensitive to noise and target size, and it only produces better segmentation results for images where the interclass variance is a single peak. The results of the segmentation using Otsu thresholding in this paper are shown in Figure 7.

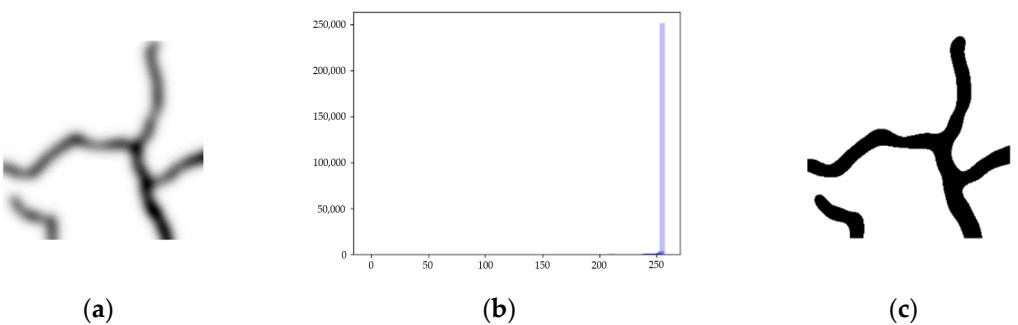

| (a) | (b) | (c) |

**Figure 7.** Otsu threshold segmentation results. (**a**) Image after Gaussian blur, (**b**) threshold distribution of images, (**c**) Otsu image after threshold segmentation.

(4)  Morphological fracture skeletonization

Crack skeletonization is the acquisition of the central axis of the image, and it uses a morphological approach to change the width of the crack image to a unit pixel value by a limited number of open operations and erosion operations. The formula is as follows.

$$A \cdot B = (A \ominus B) \oplus B \tag{8}$$

In the equation, $A \ominus B$ represents the erosion operation on image A with convolution kernel B; $\oplus$ represents the expansion operation. The results of the skeletonization of the central axis are shown in Figure 8.

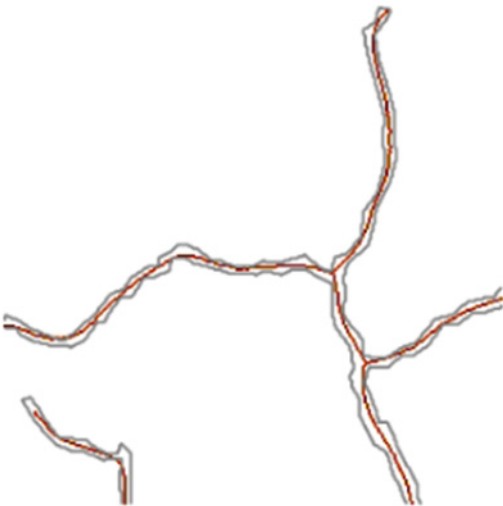

**Figure 8.** Crack skeletonization.

(5) Calculation of fracture geometry parameters.

① Calculation of crack length.

The smooth skeleton diagram is obtained after processing by the previous four algorithms, and the straight mode instead of curved was used, as the cracks have a curved course. The principle is to calculate the length of each segment in the skeleton diagram and add them up; the result is then the crack length. The crack length calculation schematic is shown in Figure 9.

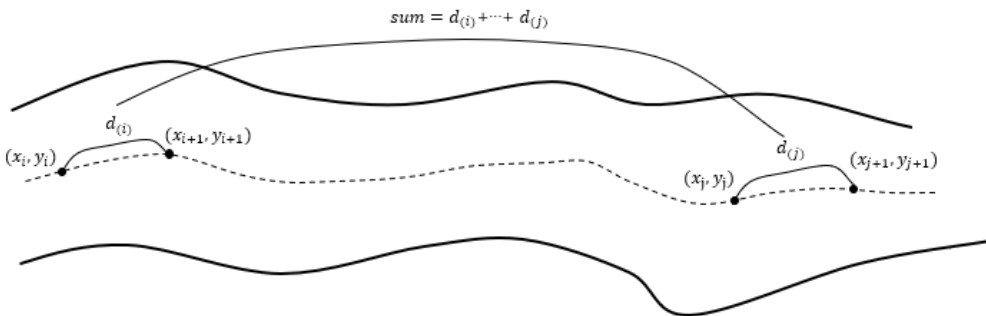

**Figure 9.** The crack length calculation schematic.

The calculation process is as follows:

(1) Iterate through the debranching skeleton to obtain the coordinates of the n sets of target points between the start and end points $(x_i, y_i)$, $i = 1, 2, \ldots, n$;

(2) Calculate the straight-line distance between adjacent points. The formula is as follows.

$$d_i = \sqrt{(x_{i+1} - x_i)^2 + (y_{i+1} - y_i)^2}, i = 1, 2, \ldots, n; \tag{9}$$

(3) Add up the straight-line distance each time:

$$\text{sum} = \sum_{i=1}^{n-1} d_i \tag{10}$$

(4) Continue the above steps until the end of the calculation of the distance between the last two points.

② Calculation of the maximum width of cracks.

Find a point in the crack skeleton, draw the normal using the direction of the tangent to that point as the direction of the pixel, find a point on the central axis that intersects the normal and calculate the distance between that point and the edge point, twice the value of which is the crack pixel width. The width of each crack in the skeleton diagram is calculated and compared to the maximum value, the result of which is the maximum crack width. The maximum width of the crack is calculated as shown in Figure 10.

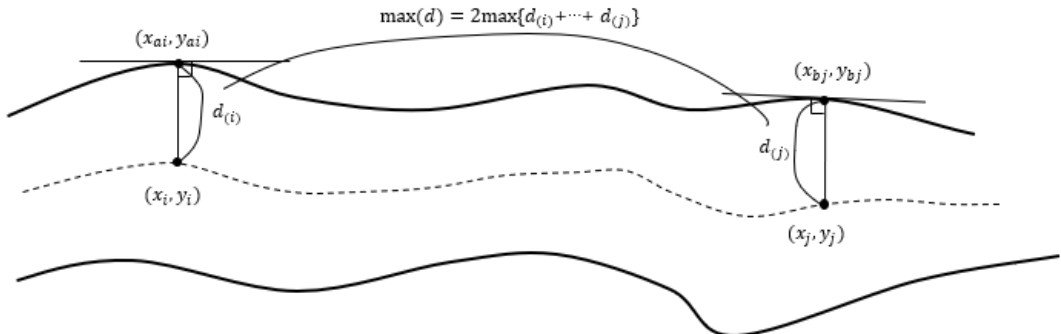

**Figure 10.** Maximum crack width calculation diagram.

The calculation process is as follows:

(1) Iterate through the debranching skeleton to obtain the coordinates of the n sets of target points between the start and end points $(x_{ai}, y_{bi}), i = 1, 2, \ldots, n; a, b = 1, 2, \ldots, n.$

(2) From the coordinates of the points on the skeleton, the orientation of the skeleton can be obtained—i.e., its normal can be determined—and according to the method described above, the coordinates of the corresponding points on the central axis can be found. The coordinates of the target point are $(x_i, y_i), i = 1, 2, \ldots, n.$

(3) At this point, twice the distance between the two points is the width of the crack:

$$d_i = \sqrt{(x_{ai} - x_i)^2 + (y_{bi} - y_{bi})^2}, i = 1, 2, \ldots, n; a, b = 1, 2, \ldots, n \tag{11}$$

(4) Compare the maximum crack width at each location:

$$\max(d) = 2\max\{d_{(i)} + \ldots + d_{(j)}\} \tag{12}$$

(5) Repeat until the width of the crack at the last point has been calculated.

The maximum widths of the cracks are visualized in Figure 11.

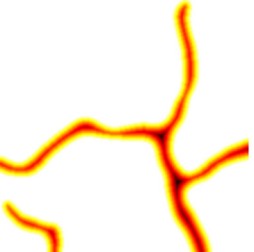

**Figure 11.** Maximum width visualization.

## 4. Model Training

Experiments were carried out on the SDNET2018 crack dataset in order to evaluate the detection performance of the proposed attention mechanism and crack geometry parameter calculation method [34]. Specifically, we first present the experimental setup and the algorithm evaluation metrics. The performances of our proposed module and network

variants are then discussed. Finally, we compare their performances with some standard methods. The configuration of the experimental environment in this paper is shown in Table 1 below.

**Table 1.** Experimental environment and configuration.

| Environment | Configuration |
|---|---|
| Operating system | Windows10 |
| CPU | Intel i5 12400F @ 2.5 GHz |
| GPU | Nvidia GeForce RTX3060 |
| RAM | 16 G |
| Memory | 500 G |
| Programming language | Python3.8 |
| Deep learning framework | Pytorch |

*4.1. Datasets*

We used the SDNET2018 crack dataset as a training sample. (SDNET2018 is an annotated image dataset for training, validation and benchmarking of artificial-intelligence-based crack detection algorithms for concrete. SDNET2018 contains over 56,000 images of cracked and non-cracked concrete bridge decks, walls and pavements. The dataset includes cracks as narrow as 0.06 mm and as wide as 25 mm. The dataset also includes images with a variety of obstructions, including shadows, surface roughness, scaling, edges, holes and background debris.) Each image was annotated using Labelme to mark the areas where the cracks are located. Figure 12a–f show the original and manually labelled images at different locations. During training, all layers of the entire network were tuned using the RMSprop algorithm to optimise the problem of excessive oscillations in the loss function in updates and to further accelerate the convergence of the function. The learning rate was set to $1 \times 10^{-4}$, and the number of training sessions (epoch) was set to 100. The training process used a migration learning method, using the pre-trained network as the initial model and setting training parameters such as learning rate, weight decay and number of iterations to construct the crack recognition model. The high-definition images (test set images) collected from the field were imported into the crack recognition model to locate the cracks in the images.

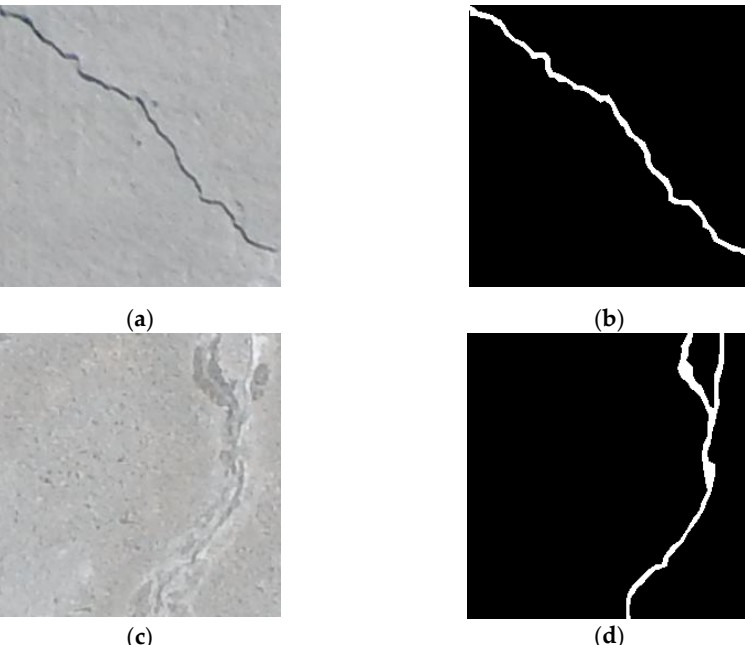

(**a**)　　　　　　　　　　　　(**b**)

(**c**)　　　　　　　　　　　　(**d**)

**Figure 12.** *Cont.*

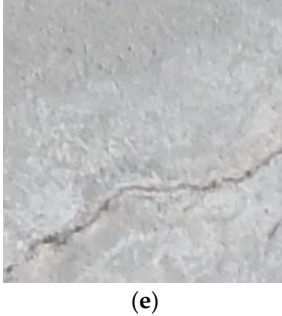
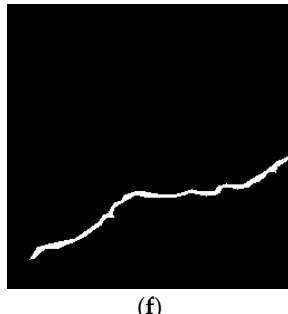

<center>(<b>e</b>)            (<b>f</b>)</center>

**Figure 12.** Original and manually labelled images. (**a**) Image I; (**b**) labelled image I; (**c**) image II; (**d**) labelled image II; (**e**) image III; (**f**) labelled image III.

### 4.2. Loss Function

Due to the superior performance of the cross-entropy loss function, the cross-entropy loss function is usually chosen as the loss function for multi-classification tasks. The cross-entropy loss function compares the prediction class and the target class for each pixel point, which is denoted as:

$$L = -\frac{1}{N} \sum_i \sum_{c=1}^{M} y_{ic} \log(p_{ic}),$$
(13)

In the formula, M denotes the number of classes. $y_{ic}$ is the sign function (0 or 1): $y_{ic}$ takes 1 when the true class of sample is equal to c; otherwise, it takes 0. $p_{ic}$ is the predicted probability that the observed sample i belongs to class c.

### 4.3. Evaluation Indicators

The model was evaluated by four metrics: the pixel accuracy (PA), category pixel accuracy (CPA), recall and IoU. The specific calculation formula is shown in Table 2. PA can be used to represent the accuracy of the model, i.e., the number of correct pixels identified by the model as a proportion of the total number of pixels. CPA represents the proportion of true positive samples among the samples identified as positive by the model. IoU is the ratio of the intersection of true and predicted values to the concatenation of true and predicted values. TP represents the number of pixels correctly identified as positive, TN represents the number of pixels correctly identified as negative, FP represents the number of pixels identified as positive that are in fact negative, and FN represents the number of pixels of negative samples identified as positive.

**Table 2.** Calculation formula for evaluation indicators.

| Evaluation Index | Computational Formula |
|:---:|:---:|
| PA | $\frac{TP+TN}{TP+TN+FP+FN}$ |
| CPA | $\frac{TP}{TP+FP}$ |
| Recall | $\frac{TP}{TP+FN}$ |
| IoU | $\frac{TP}{TP+FP+FN}$ |

### 4.4. Evaluation Indicators

To verify the feasibility of the CBAM attention mechanism, a comparison between the method of this paper (CBAM-Unet) and U-net was carried out. As can be seen in Figure 13, both methods can detect crack locations, but CBAM-Unet has higher detection accuracy. To further verify the applicability of the algorithm, experiments were conducted under dark conditions (Figure 14) and fine crack conditions (Figure 15). Based on the experimental results, it can be concluded that the CBAM-Unet algorithm in this paper, which adds an attention mechanism to identify the detailed information of cracks, is more accurate.

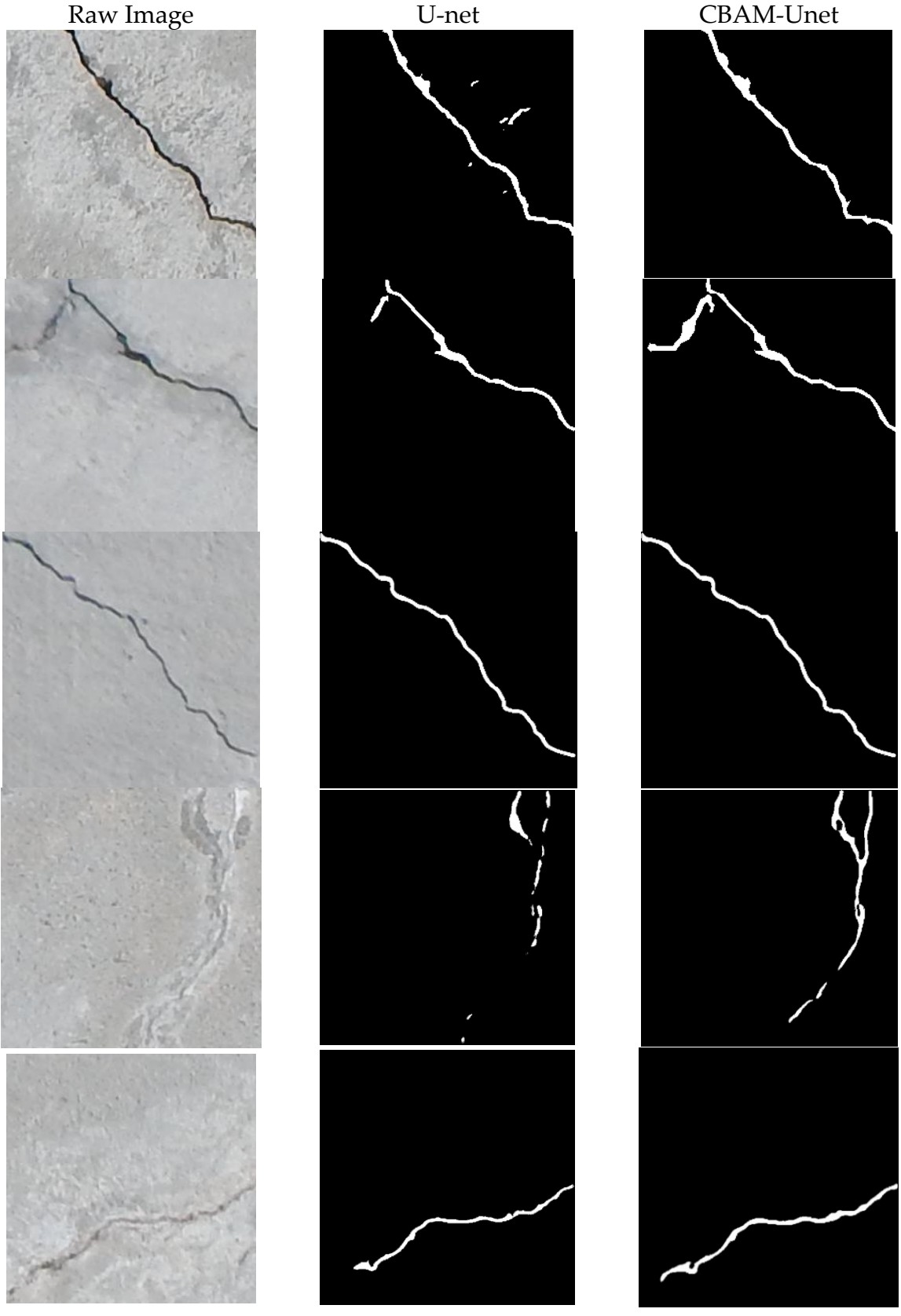

**Figure 13.** Results of the two methods under ideal conditions.

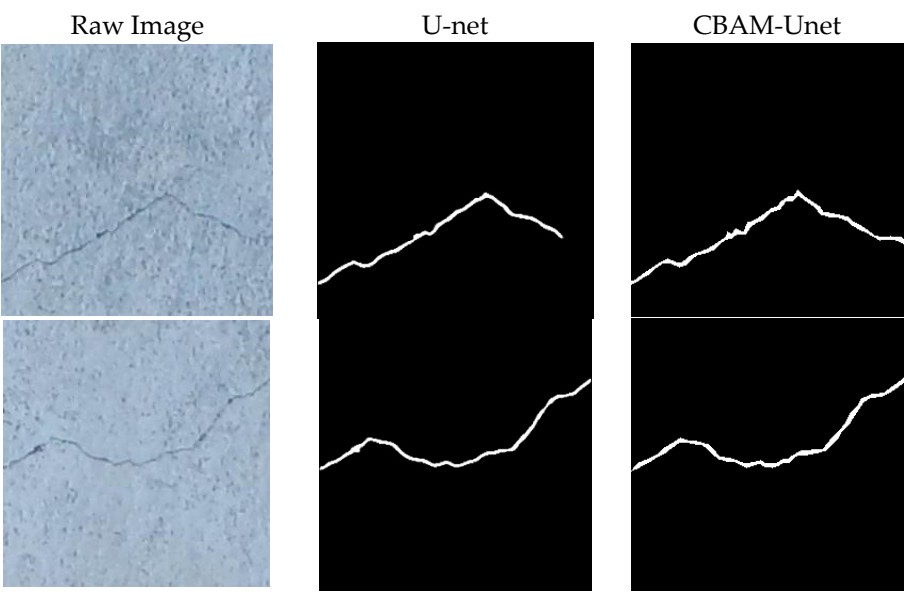

**Figure 14.** Results of the two methods in dark conditions.

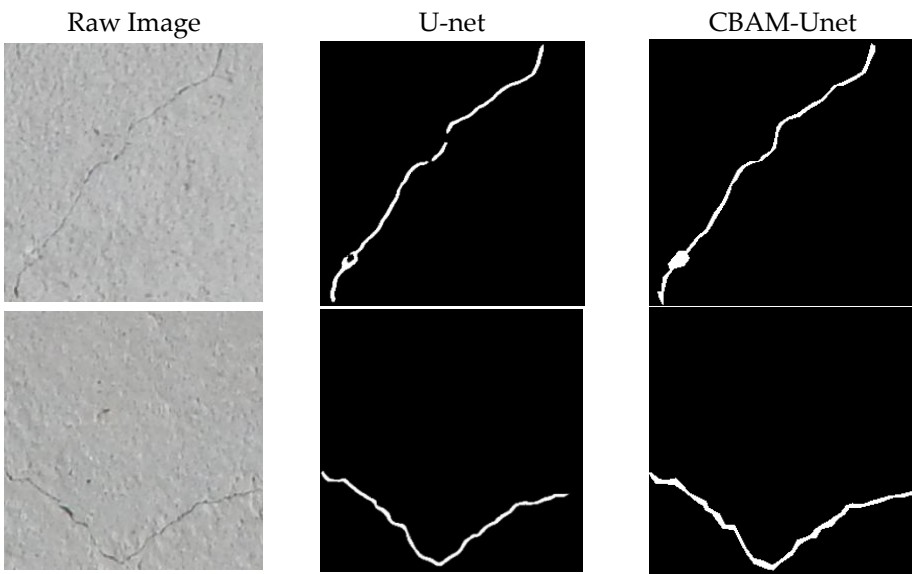

**Figure 15.** Results of two methods for shallow cracks.

According to the above experiments, CBAM-Unet has the highest detection accuracy in different environments, and it has higher robustness. The experimental comparison results are shown in Table 3.

**Table 3.** Accuracy of crack detection.

| Methods | PA | CPA | Recall | IoU |
|---------|------|------|--------|------|
| U-net | 87.32% | 84.96% | 96.26% | 82.16% |
| CBAM-Unet | 92.66% | 92.20% | 97.13% | 89.53% |

On the basis of U-net, the channel and spatial attention modules improve the representation ability of network feature extraction. Compared with the reference network, the accuracy of the improved network has been significantly improved.

## 5. Verification Experiments on the Accuracy of Calculating Crack Geometry Parameters

To verify the accuracy of the orthogonal skeleton line method for extracting crack geometry parameters, the output of the binary images from the CBAM-Unet algorithm in Section 4 of this paper was calculated and compared with that of the manual detection method.

The experimental object was a cable tower of a cross-sea bridge in Qingdao, China, and a high-definition camera was used for image acquisition, as shown in Figure 16. In order to improve the detection accuracy and speed, we pre-processed the collected image and divided it into several 256 ∗ 256 pictures, making it the same format as the dataset used in this paper.

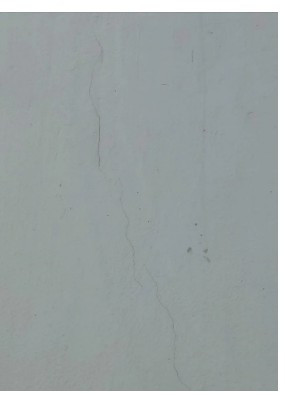 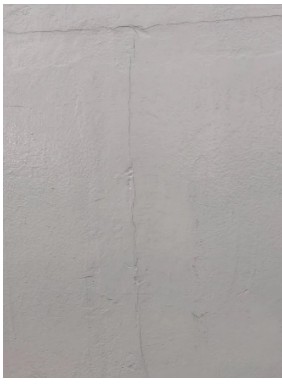

**Figure 16.** Images of bridge towers.

### 5.1. Pixel Calibration

The physical size corresponding to each pixel in the image was calculated through pixel calibration, and the physical size can be calculated through the camera pinhole model.

$$L_{PP} = \frac{10L_d}{L_f P_C} \tag{14}$$

In the formula, $L_{PP}$ is the physical size corresponding to each pixel, $L_d$ is the shooting distance, $L_f$ is the focal length of the camera, and $P_C$ is the number of pixels corresponding to the camera's photosensitive element of 1 cm.

### 5.2. Experimental Results

The manual width measurement method uses a crack width tester and a crack detector, and the length is measured by fitting a thin line to the crack alignment, straightening it and then using a ruler to measure the length of the thin line. The results of the comparison are shown in Tables 4 and 5.

**Table 4.** Comparison of measured values of maximum crack widths.

| Number | Crack Width (mm) | | Inaccuracy | |
|---|---|---|---|---|
| | Calculated Values (mm) | Measured Values (mm) | Absolute Values/mm | Relative Values/% |
| 1 | 0.612 | 0.630 | −0.018 | 2.9 |
| 2 | 0.962 | 0.980 | −0.018 | 1.8 |
| 3 | 1.448 | 1.360 | 0.088 | 6.5 |
| 4 | 1.560 | 1.620 | −0.060 | 3.7 |
| 5 | 1.208 | 1.260 | −0.052 | 4.1 |
| 6 | 0.826 | 0.840 | −0.014 | 1.7 |
| 7 | 2.244 | 2.200 | 0.044 | 2.0 |
| 8 | 1.762 | 1.720 | 0.042 | 2.4 |
| 9 | 1.706 | 1.620 | 0.086 | 5.3 |
| 10 | 2.248 | 2.160 | 0.088 | 4.1 |

**Table 5.** Comparison of crack length measurements.

| Number | Crack Length (mm) | | Inaccuracy | |
|---|---|---|---|---|
| | Calculated Values (mm) | Measured Values (mm) | Absolute Values/mm | Relative Values/% |
| 1 | 39.308 | 39.940 | −0.632 | 1.58 |
| 2 | 36.600 | 38.368 | −1.768 | 4.61 |
| 3 | 38.256 | 39.572 | −1.316 | 3.33 |
| 4 | 36.272 | 38.224 | −1.952 | 5.11 |
| 5 | 38.400 | 39.658 | −1.258 | 3.17 |
| 6 | 32.068 | 34.720 | −2.652 | 7.64 |
| 7 | 34.496 | 35.970 | −1.474 | 4.10 |
| 8 | 37.568 | 35.980 | 1.588 | 4.41 |
| 9 | 34.884 | 36.234 | −1.350 | 3.73 |
| 10 | 36.544 | 37.896 | −1.352 | 3.57 |

Each pixel corresponds to physical size of 0.293 mm.

Table 4 shows that the absolute error of the maximum crack width of the regular sample is small and meets the detection requirements. However, the smaller the crack width, the larger the relative error, which may be due to the inaccuracy of the calculation results due to the lack of accuracy of the measurement results when performing manual measurements or the misprocessing of some crack information during image processing.

From Table 5, we can see that the relative error data of crack length is distributed in the range of 1% to 8%—relatively high accuracy—but some of the data show large errors, which were caused by the loss of some information when denoising the crack images.

## 6. Conclusions

The main aim of this study was to improve the expression of shallow features in crack images and to improve the crack segmentation effect. Based on the principle of image detection, the CBAM-Unet bridge crack segmentation algorithm was proposed, which contains a dual attention mechanism. The bridge crack image is extracted by an extraction algorithm, an image processing algorithm, and a geometric parameter measurement algorithm to obtain the binary image, maximum width and length of the crack, and other parameters to meet the needs of engineering inspection. In this study, the channel attention module and the spatial attention module were added to the max-pooling process of the U-net network. Spatial attention allows the neural network to focus more on the pixel regions of the image that are decisive for classification and ignore the insignificant regions, and channel attention is used to deal with the assignment relationship of the feature map channels. The experimental results show that CBAM-Unet is more accurate than U-net for bridge crack segmentation, having an accuracy of 92.66%. The maximum crack width error calculated based on this method was between 1% and 6%, and the crack length error was between 1% and 8%, which meets the detection requirements.

There are certain limitations to this paper. We only extracted features from cracks in bridges and did not automatically count the cracks, and the impact of cracks on bridges needs further evaluation. Therefore, a future research direction could be to perform an automatic evaluation based on the extracted cracks, so as to accurately obtain the true state of the bridge and provide a basis for decision-making for maintenance personnel. In addition, the detection of other structural defects in bridges is also an important research direction, which is of great significance for bridge maintenance.

**Author Contributions:** Conceptualization, H.S.; methodology, X.W.; software, X.W.; validation, X.W. and Z.W.; formal analysis, Z.Z.; investigation, X.W. and P.Z.; resource, P.Z. and Z.Z.; data curation, T.H.; writing—original draft preparation, H.S.; writing—review and editing, X.W., Z.W., P.Z. and Z.Z. All authors have read and agreed to the published version of the manuscript.

**Funding:** This project is supported jointly by the "Assessment of concrete crack condition and repair technology for cross-sea cable-stayed bridges and suspension bridges in the northern frozen sea area (2022QDFZYG02)" and the "Science and Technology Plan of Shandong High Speed Group Co (HSB2020117)".

**Data Availability Statement:** Not applicable.

**Conflicts of Interest:** The authors declare no conflict of interest.

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
