# Peer review of "Research on a U-Net Bridge Crack Identification and Feature-Calculation Methods Based on a CBAM Attention Mechanism"

_buildings, doi:10.3390/buildings12101561_

Round 1

Reviewer 1 Report

In this paper, the bridge crack identification algorithm for Unet that combines the channel attention module (CAM) and spatial attention module (SAM), called the CBAM Unet algorithm was proposed. This algorithm could achieve an accuracy of 92.66% for crack identification and could complete the measurement of the maximum width and length of the crack with errors between 1%-6% and 1%-8% respectively. This paper provides beneficial findings and is judged to be an excellent paper worthy of publication.

However, please consider correcting or adding to the following items.

1.       Does ``Traditional bridge crack detection'' mean visual inspection? Could you please describe the conventional method in detail? (Line 48)

2.       Please correct 2.1. to 2.2. (Line 132)

3.       Please explain "CRACK500" concretely. I can't understand what you mean by just showing the references. (Line 262)

4.       Please explain ”SDNET2018” concretely. (Line 271)

5.       It is easier to understand if you clearly indicate what TP, TN, FP, and FN represent. (Table 2)

Author Response

1.“传统桥梁裂纹检测”是指目视检查吗?您能详细描述一下传统方法吗?(第48行)

响应 1:感谢您的提问,对于传统的桥梁裂缝检测,说明如下,传统的桥梁裂缝检测方法主要有以下两种:首先要求检查员事先设置一个工作平台,然后用肉眼或其他设备进行观察和检测;第二种方法是通过桥梁检查车辆的篮子或桁架装置将检查员和设备送到桥顶进行观察。

2.请更正 2.1.更改为 2.2。(第132行)

响应 2: 感谢您指出此错误,其中2.1已更改为2.2。

3.请具体说明“裂纹500”。我不明白你仅仅显示参考文献是什么意思。(262线)

回应3:谢谢你的问题,我已经将CRACK500更改为SDNET2018,这两个数据集都是开源的混凝土开裂数据集,我们最终选择使用SDNET2018。

4.请具体说明“SDNET2018”。(271线)

响应4:SDNET2018是一个带注释的图像数据集,用于训练,验证和基准测试基于人工智能的混凝土裂缝检测算法。SDNET2018 包含超过 56,000 张开裂和未开裂混凝土桥面、墙壁和路面的图像。数据集包括窄至 0.06 mm 和宽至 25 mm 的裂缝。该数据集还包括具有各种障碍物的图像,包括阴影、表面粗糙度、缩放、边缘、孔洞和背景碎片。我在案文第299至305行中加上了上述内容。

如果您清楚地指明 TP、TN、FP 和 FN 代表什么,则 5.It 更容易理解。(表2)

答复5:谢谢你的建议,我在论文第333至336行详细描述了这些建议。“其中TP表示识别的阳性样品的像素数,TN表示将阳性样品识别为负样本的像素数,FP表示从负样本中识别阳性样品的像素数,FN表示识别的负样本的像素数。

Reviewer 2 Report

Paper Review

Title: “Research on U–Net Bridge Crack Identification and Feature Calculation Method Based on CBAM Attention Mechanism”

Authors: H. Su, X. Wang, T. Han, Z. Wang, Z. Zhao, P. Zhang

Manuscript ID: 1903919

Journal: Buildings

Review Comments

This paper presents a bridge crack detection algorithm based on U–net and combining channel attention module (CAM) and spatial attention module (SAM). There are some critical issues and concerns, and the authors should address the following comments.

Comment 1.                   The English of the paper needs to be improved. There are some syntax and grammar errors.

Comment 2.                   Heading numbering is wrong. In Section “2. U–net methods and channels, spatial attention mechanisms”, the second subsection must be “2.2. Design of CBAM–U–net method based on attention mechanism”.

Comment 3.                   Equation numbering is wrong as well.

Comment 4.                   In Abstract, what does “CBAM” stand for?

Comment 5.                   In line 162, it must be MS.

Comment 6.                   In Eq. (3), what does “MLP” stand for?

Comment 7.                   Please show the image processing algorithm in Figure 5 as a flow chart, and then explain each step.

Comment 8.                   Step “(5) Calculation of fracture geometry parameters” needs to be more discussed.

Comment 9.                   Please provide the outputs of U–net and CBAM–Unet for the images given in Figure 10.

Best Regards,

Author Response

Comment 1. The English of the paper needs to be improved. There are some syntax and grammar errors.

We have undergone English language editing by MDPI. The text has been checked for correct use of grammar and common technical terms, and edited to a level suitable for reporting research in a scholarly journal.

Comment 2. Heading numbering is wrong. In Section “2. U–net methods and channels, spatial attention mechanisms”, the second subsection must be “2.2. Design of CBAM–U–net method based on attention mechanism”.

Response 2: Thank you for pointing out the error, I have amended the second heading to 2.2.

Comment 3. Equation numbering is wrong as well.

Response 3: Thank you for pointing out the equation numbering error, I have corrected the equation numbering.

Comment 4. In Abstract, what does “CBAM” stand for?

Response 4: CBAM (Convolutional Block Attention Module) is a lightweight convolutional attention module that combines channel and spatial attention mechanism modules. CBAM contains two sub-modules, Channel Attention Module and Spatial Attention Module, which perform Attention on channel and spatial respectively.

Comment 5. In line 162, it must be MS.

Response 5: Thank you for pointing out this error, I have changed  to .

Comment 6. In Eq. (3), what does “MLP” stand for?

Response 6: MLP stands for the Share MLP module in the channel attention module. In this module the number of channels is first compressed and then expanded to the original number of channels, and the result of the two activations is obtained by the ReLU activation function.

Comment 7. Please show the image processing algorithm in Figure 5 as a flow chart, and then explain each step.

Response 7: We strongly agree with your suggestion and we have changed Figure 5 into a flowchart and (1) to (5) explain the algorithms in Figure 5 respectively.

Comment 8. Step “(5) Calculation of fracture geometry parameters” needs to be more discussed.

Response 8: We strongly agree with you that we have added to the original for the crack length and maximum width by adding an algorithm schematic and algorithm flow.

Comment 9. Please provide the outputs of U–net and CBAM–Unet for the images given in Figure 10.

Response 9: Thank you for pointing out the deficiencies, as due to a change in space, the original Figure 10 has now become Figure 12 and is hereby noted. We have included the output of the U-net and CBAM-Unet for the image shown in Figure 12 in Figure 13.

Round 2

Reviewer 2 Report

Paper Review

Title: “Research on U–Net Bridge Crack Identification and Feature Calculation Method Based on CBAM Attention Mechanism”

Authors: H. Su, X. Wang, T. Han, Z. Wang, Z. Zhao, P. Zhang

Manuscript ID: buildings–1903919

Journal: Buildings

Review Comments

Regarding the above submission, the authors have addressed most of my comments in the previous round of review, but there are still some minor issues and concerns.

Comment 1.                   Your response to Comment 4 of previous review [Comment 4. In Abstract, what does “CBAM” stand for?] should be added to Abstract.

Comment 2.                   Your response to Comment 6 of previous review [Comment 6. In Eq. (3), what does “MLP” stand for?] should be added to the manuscript in a suitable place.

Best Regards,

Author Response

Comment 1. Your response to Comment 4 of previous review [Comment 4. In Abstract, what does “CBAM” stand for?] should be added to Abstract.

Thanks for the correction, we have added what CBAM stands for to the summary.

Comment 2. Your response to Comment 6 of previous review [Comment 6. In Eq. (3), what does “MLP” stand for?] should be added to the manuscript in a suitable place.

Thanks for the correction, we have added the meaning of MLP to the paper.